# Delineating Zinc Influx Mechanisms during Platelet Activation

**DOI:** 10.3390/ijms241411689

**Published:** 2023-07-20

**Authors:** Sahithi J. Kuravi, Niaz S. Ahmed, Kirk A. Taylor, Emily M. Capes, Alex Bye, Amanda J. Unsworth, Jonathan M. Gibbins, Nicholas Pugh

**Affiliations:** 1School of Life Sciences, Anglia Ruskin University, Cambridge CB1 1PT, UKemc166@pgr.aru.ac.uk (E.M.C.); 2Institute for Cardiovascular and Metabolic Research, School of Biological Sciences, University of Reading, Reading RG6 6EX, UKj.m.gibbins@reading.ac.uk (J.M.G.); 3Department of Life Sciences, Faculty of Science and Engineering, Manchester Metropolitan University, Manchester M1 5GD, UK

**Keywords:** zinc, zinc entry, platelets, zinc-induced platelet activation, aggregation, cation signaling, TRP channels, NCX, ZIP7, store-operated calcium entry, store-operated zinc entry

## Abstract

Zinc (Zn^2+^) is released by platelets during a hemostatic response to injury. Extracellular zinc ([Zn^2+^]_o_) initiates platelet activation following influx into the platelet cytosol. However, the mechanisms that permit Zn^2+^ influx are unknown. Fluctuations in intracellular zinc ([Zn^2+^]_i_) were measured in fluozin-3-loaded platelets using fluorometry and flow cytometry. Platelet activation was assessed using light transmission aggregometry. The detection of phosphoproteins was performed by Western blotting. [Zn^2+^]_o_ influx and subsequent platelet activation were abrogated by blocking the sodium/calcium exchanged, TRP channels, and ZIP7. Cation store depletion regulated Zn^2+^ influx. [Zn^2+^]_o_ stimulation resulted in the phosphorylation of PKC substates, MLC, and β3 integrin. Platelet activation via GPVI or Zn^2+^ resulted in ZIP7 phosphorylation in a casein kinase 2-dependent manner and initiated elevations of [Zn^2+^]_i_ that were sensitive to the inhibition of Orai1, ZIP7, or IP_3_R-mediated pathways. These data indicate that platelets detect and respond to changes in [Zn^2+^]_o_ via influx into the cytosol through TRP channels and the NCX exchanger. Platelet activation results in the externalization of ZIP7, which further regulates Zn^2+^ influx. Increases in [Zn^2+^]_i_ contribute to the activation of cation-dependent enzymes. Sensitivity of Zn^2+^ influx to thapsigargin indicates a store-operated pathway that we term store-operated Zn^2+^ entry (SOZE). These mechanisms may affect platelet behavior during thrombosis and hemostasis.

## 1. Introduction

Dietary zinc (Zn^2+^) deficiency results in prolonged bleeding phenotypes in both rodents [1,2,3] and humans [4,5,6] that can be reversed by dietary supplementation. Plasma Zn^2+^ concentrations range from 10 to 20 µM; however, binding to plasma proteins (i.e., albumin and α_2_ microglobulin) reduces the free (labile) concentration to around 0.5 µM [7,8,9]. Zn^2+^ concentrations are elevated within atherosclerotic plaques and at sites of vascular injury as a result of a release from platelet granules, and are likely to be much higher than the average plasma concentration [10,11].

At concentrations of 100 μM, Zn^2+^ acts as a platelet agonist, regulating intracellular signaling resulting in granule secretion, the tyrosine phosphorylation of protein kinase C (PKC), integrin α_IIb_β_3_ activation, and platelet aggregation [12,13,14]. At low concentrations, Zn^2+^ is a coactivator, potentiating platelet activation to threshold concentrations of conventional agonists [12,15]. Previous research has demonstrated changes in platelet intracellular Zn^2+^ concentrations [Zn^2+^]_i_ following agonist stimulation, indicating a release from intracellular stores, which is consistent the a role of Zn^2+^ as an intracellular secondary messenger [14]. Agonist-evoked increases in [Zn^2+^]_i_ in the absence of [Zn^2+^]_o_ result in platelet activation, shape change, degranulation, and phosphatidyl-serine exposure. Similarly, Zn^2+^ regulates Ca^2+^ release from stores and may therefore be a central factor in regulating platelet activation [16,17].

In nucleated cells, Zn^2+^ permeability across the cell membrane is regulated by Zn^2+^ transporting proteins, including non-selective cation channels, transporters, and exchangers [18,19,20]. Cellular Zn^2+^ homeostasis is regulated by Zn^2+^ transporters (ZnTs) and Zrt-Irt-like proteins (ZIPs), of which transcripts are detected in megakaryocytes [20,21,22]. Although the Na^2+^/Ca^2+^ exchanger (NCX), operating in reverse mode, and transient receptor potential (TRP) channels have been reported to facilitate Zn^2+^ movement in nucleated cells, their role in platelet Zn^2+^ homeostasis has yet to be investigated [23,24]. [Zn^2+^]_i_ buffering systems are important mechanisms for regulating Zn^2+^ bioavailability [25]. In platelets, agonist-evoked increases in [Zn^2+^]_i_ are regulated by changes in the platelet redox state, suggestive of a role for redox-sensitive proteins, such as metallothioneins [26].

Intracellular platelet calcium concentrations ([Ca^2+^]_i_) are maintained by store-operated Ca^2+^ entry (SOCE) and receptor-operated Ca^2+^ entry (ROCE) pathways, where the dense tubular system (DTS) is the primary Ca^2+^ store [27,28,29]. SOCE is induced via the activation of cation-selective CRAC channels [30,31,32,33]. The resting platelet Ca^2+^ concentration ([Ca^2+^]_rest_) is maintained at approximately 100 nM by the action of channels and exchangers expressed along the plasma membrane and the surface of the DTS [34,35]. Interestingly, ryanodine receptors, expressed along the cardiomyocyte sarcoplasmic reticulum, have been shown to be regulated by the elevation of [Zn^2+^]_i_; although, it is not clear whether the equivalent IP_3_R in platelets is modulated in a similar manner [36,37].

We have previously shown that the exposure of platelets to [Zn^2+^]_o_ leads to its rapid and sustained accumulation within the platelet cytosol [14,26]. In this study, we investigate the mechanisms underpinning platelet Zn^2+^ influx, and the storage mechanisms that contribute to Zn^2+^-induced platelet activation (ZIPA).

## 2. Results

### 2.1. TRP Channels and Reverse-Mode NCX Regulate Zn^2+^ Influx during Zn^2+^-Induced Platelet Activation

Zn^2+^ acts in a dual manner to regulate platelet activation. When applied extracellularly, Zn^2+^ gains access to the platelet cytosol, whilst, in the absence of extracellular Zn^2+^, platelet activation via canonical receptors, such as GPVI, results in increases in intracellular Zn^2+^ as a result of a release from stores [12,14,26]. Increases in [Zn^2+^]_i_ via either pathway results in ZIPA, which is sensitive to a range of inhibitors and metal ion chelators [12]. However, the molecular identity and the contribution of cation channels underlying Zn^2+^ influx in platelets have not been investigated. TRP proteins are non-selective cation channel of which, TRPC6, TRPC1, TRPV1, and TRPC5 are expressed in platelets or megakaryocytes, and have been reported to facilitate cellular Zn^2+^ uptake in nucleated cells [24,38,39,40]. Additionally, the Na^+^/Ca^2+^ exchanger (NCX) substitutes Ca^2+^ for Zn^2+^ in order to mediate Zn^2+^ influx in intestinal epithelial cells [18,41].

We evaluated the roles of TRP channels and NCX during ZIPA. Platelets were stimulated by exposure to 100 µM ZnSO_4_ in the presence of increasing concentrations of TRP channel blockers 2-APB (Figure 1a), FFA (Figure 1b), and SKF (Figure 1d), or KB-R7943 mesylate, which is a reverse-mode NCX inhibitor and has also been shown to be a potent blocker of TRP channels (KB-R, Figure 1c) [42]. ZIPA was reduced in a concentration-dependent manner following treatment with each inhibitor. Calculated IC_50_ values (Figure 1e, Table 1) were comparable for 2-APB, KB-R, and SKF, whilst the curve for FFA was right-shifted. Platelet aggregation responses to 100 µM Zn^2+^ were reduced from 73.9 ± 12.1% in the presence of KB-R alone, and to 10.4 ± 5.2% by the combined blocking of KB-R and 2-APB (Figure 1f, *p* < 0.01). Residual ZIPA responses suggested that multiple Zn^2+^-permeable channels, and potentially Zn^2+^ transporters, contributed to platelet responses to Zn^2+^.

The stimulation of platelets with [Zn^2+^]_o_ leads to a concentration-dependent increase in [Zn^2+^]_i_; however, the nature of the Zn^2+^ influx pathway or the relative contribution of the release of [Zn^2+^]_i_ stores remains unknown [12]. We assessed the role of NCX and TRP channels in platelet Zn^2+^ influx using FZ-3-loaded platelets. Platelets were stimulated by 100 µM Zn^2+^ following treatment with inhibitors, KB-R or SN-6, or 2-APB and the resultant Zn^2+^ signals were quantified using flow cytometry (Figure 2a). An inhibition of Zn^2+^ influx was observed upon pre-incubation with 30 µM KB-R (to 2611 ± 516 a.u.) or 0.5 µM SN-6, (to 2537 ± 204 a.u.) compared with 4882 ± 531 a.u. for the vehicle control, representing decreases of 53.4 ± 3.2% and 51.9 ± 1.8%, respectively (Figure 2a, *p* < 0.01). The pre-incubation of FZ-3-loaded platelets with 2-APB reduced the peak Zn^2+^ influx to 2035 ± 322 a.u. (41.7 ± 3.6% of vehicle control, Figure 2a, *p* < 0.001), and to 2662 + 420 a.u. following co-treatment with 30 µM KB-R and 2-APB, as observed in the platelet aggregation (54.5 ± 3.6% of vehicle control, Figure 2a, *p* < 0.001). These data are suggestive of a constitutively active Zn^2+^ entry pathway that allows platelets to sense changes in local Zn^2+^ concentrations.

As KB-R is an inhibitor of both NCX and TRP channels, we further investigated the roles of specific TRP channels in Zn^2+^ influx [42]. A range of TRP inhibitors were screened for their influence on the stimulation of FZ-3-loaded platelet suspensions with 100 μM Zn^2+^. Peak Zn^2+^ influx was reduced from 2612 ± 198 a.u (for the vehicle control) to 643 ± 103.9 a.u. in platelets treated with the TRPC6 inhibitor SAR7334 (Figure 2b, *p* < 0.001). Similarly, treatment with TRPC5 or TRPV1 blockers (AC1903 and AMG9810, respectively) reduced Zn^2+^ influx to 711 ± 136 and 1328 ± 173 a.u., respectively (Figure 2b, *p* < 0.001). The inhibition of TRPs that were not expressed on platelets (TRPM4 and TRPA1) [43,44] with AM0902 or 9-Phenanthrol, respectively, had no effect on Zn^2+^ influx (mean fluorescence intensities were 2462 ± 248 and 2766 ± 439 a.u. respectively; Figure 2b, ns). These data indicate that a component of observed [Zn^2+^]_o_ influx was insensitive to NCX and TRP blockers, suggesting that other Zn^2+^ entry pathways may also contribute to Zn^2+^ influx. Such pathways can include Zn^2+^ transporters or the release of Zn^2+^ into the cytosol from internal stores.

Further experiments were performed to further confirm the role of TRP channels in [Zn^2+^]_o_ influx. OAG and the PKC inhibitor, GO6983, have previously been used to potentiate TRPC6 activity in HEK293 and smooth muscle cells [45,46]. FZ-3-loaded platelets were pre-treated with OAG, GO6983, or a combination of the two prior to stimulation by Zn^2+^. Zn^2+^ influx was increased in dual-treated platelets (F/F_0_ for dual-treated platelets was 9.3 ± 0.2 a.u., compared to untreated platelets: 5.9 ± 0.4 a.u.; Figure 2c,d, *p* < 0.01.), supporting the evidence that TRP channels are a route for Zn^2+^ influx.

### 2.2. Zn^2+^-Induced Platelet Activation Is Associated with the Phosphorylation of Substrates of Ca^2+^-Dependent Enzymes

Whilst ZIPA produces a distinct pattern of tyrosine phosphorylation, the effect on other phosphoproteins has not been studied [14]. It has been reported that Zn^2+^ can substitute for Ca^2+^ at binding sites on calmodulin and other Ca^2+^-dependent enzymes [47,48]. Therefore, we investigated the association of ZIPA with the phosphorylation and activation of key activatory proteins, including the Ca^2+^-dependent myosin light chain kinase (MLCK) and PKC. Previous work using PKC inhibitors highlighted a role for MLCK and PKC during ZIPA [14], with Zn^2+^ possibly substituting for Ca^2+^ at the active site of these enzymes, leading to their activation [17]. Platelet lysates were prepared at 0, 2, 5, and 10 min intervals following stimulation by activatory (1 mM) or sub-activatory (30 µM) concentrations of Zn^2+^ (Figure 3). Increases in the protein phosphorylation of PKC and MLCK substrates alongside the phosphorylation of the integrin β3 chain occurred in a temporal manner and peaked within 5 min of stimulation with 1 mM Zn^2+^ (Figure 3), consistent with the time course for ZIPA. The phosphorylation of these substrates was observed in platelet populations treated with the integrin α_IIb_β_3_. antagonist integrilin, demonstrating that phosphorylation was not attributable to outside–in signaling through α_IIb_β_3_.

### 2.3. Zn^2+^ Influx Is Mediated by Ca^2+^ Store Depletion

Upon receptor-mediated activation, platelets generate IP_3_ that activates IP_3_R leading to the release of Ca^2+^ from intracellular stores into the cytosol. This process is supported by casein kinase 2 (CK2), which has also been shown to be involved in ZIP7 phosphorylation in nucleated cells [49,50]. SOCE or ROCE both act to increase [Ca^2+^]_o_ uptake, and we hypothesized that these mechanisms might also be responsible for Zn^2+^ influx. Washed platelet suspensions were loaded with FZ-3 or Fluo-4, and thrombin- or thapsigargin (TG)-induced fluctuations of [Ca^2+^]_i_ or [Zn^2+^]_i_ were investigated using flow cytometry (Figure 4). As shown previously, thrombin caused a rapid increase in [Ca^2+^]_i_, which was enhanced in the presence of 2 mM [Ca^2+^]_o_, consistent with SOCE (F/F_0_ peaked at 5.3 ± 0.4 a.u. compared to 4.4 ± 0.2 a.u for thrombin alone, Figure 4a,b). In the presence of subactivatory concentrations (30 μM) of [Zn^2+^]_o_, thrombin-mediated [Ca^2+^]_i_ increases were significantly reduced (to 2.7 ± 0.1 a.u., *p* < 0.001). This effect was partially restored with the addition of 2 mM [Ca^2+^]_o_ (3.2 ± 0.0 a.u., *p* < 0.001, Figure 4a,b), indicating that [Zn^2+^]_o_ has a negative effect on Ca^2+^ influx. Fluctuations in Ca^2+^ were sensitive to pre-treatment with the non-specific cation chelator BAPTA-AM (10 µM), confirming the nature of the [Ca^2+^]_i_ response. Pre-treatment with the intracellular Zn^2+^ chelator, TPEN (50 µM), had no significant effect on thrombin-induced [Ca^2+^]_i_ signals, but abolished the reductive effect of [Zn^2+^]_o_ on [Ca^2+^]_i_ signals (F/F_0_ was 4.0 ± 0.1 a.u compared to 4.4 ± 0.2 a.u; Figure 4a,b, ns), further confirming the role of Zn^2+^ in regulating platelet [Ca^2+^]_i_ responses.

As 2-APB and KB-R were implicated in Zn^2+^ influx (Figure 1), these pathways were assessed to further explore the mechanisms that contributed to the Zn^2+^ regulation of SOCE. In 2 mM [Ca^2+^]_o_, 2-APB caused a significant decrease in thrombin-evoked [Ca^2+^]_i_ responses (from 5.3 ± 0.4 a.u to 3.0 ± 0.3, *p* < 0.01; Figure 4a,b). As 2-APB affected both TRP channels and IP_3_R, these findings implicate Zn^2+^ in the regulation of cation entry via TRP channels or Ca^2+^ or Zn^2+^ release via DTS. KB-R pre-treatment had no significant effect on thrombin-evoked [Ca^2+^]_i_ fluctuations, irrespective of the cation conditions tested. Interestingly, in 2-APB-treated platelets, thrombin-evoked Ca^2+^ signals in the presence of [Zn^2+^]_o_ did not differ to the vehicle controls. Thus, 2-APB-sensitive pathways were not involved in the Zn^2+^-mediated regulation of [Ca^2+^]_i_.

To investigate whether [Zn^2+^]_o_ influenced Ca^2+^ influx following store depletion, Fluo-4-loaded platelets were stimulated by TG in the presence and absence of [Zn^2+^]_o_, and [Ca^2+^]_I_ fluctuations were quantified using flow cytometry. TG (1 µM) stimulation elevated [Ca^2+^]_i_ to 2.0 ± 0.1 a.u (Figure 4c,d), reflecting Ca^2+^ release from intracellular stores. This was increased significantly in the presence of [Ca^2+^]_o_, consistent with SOCE (peak Fluo-4 fluorescence was 4.5 + 0.5 a.u.; Figure 4c,d, *p* < 0.05). In the presence of [Zn^2+^]_o_, Ca^2+^ influx was reduced by 72% (to 1.3 + 0.1; Figure 4c,d, *p* < 0.05). In the presence of both [Ca^2+^]_o_ and [Zn^2+^]_o_, peak TG-induced [Ca^2+^]_i_ signals were reduced to 3.0 + 0.3 a.u. (Figure 4c,d, *p* < 0.01) suggesting a competition between Zn^2+^ and Ca^2+^ for cation entry mechanisms. As expected, BAPTA pre-treatment abolished Ca^2+^ responses to TG treatment, irrespective of extracellular cation status (Figure 4c,d, *p* < 0.05). The peak signal evoked by 1 μM TG upon TPEN (25 μM) pre-treatment was 1.8 ± 0.1 a.u, compared to the vehicle control (2.0 ± 0.1 a.u; Figure 4c,d, ns) indicating that increases in [Zn^2+^]_i_, either as a result of its release from stores or via Zn^2+^ entry, did not affect SOCE. KB-R pre-treatment did not affect Ca^2+^ influx, whilst 2-APB pre-treatment significantly reduced Ca^2+^ influx in the presence of 2 mM [Ca^2+^]_o_ (peak fluorescence was 3.3 + 0.4 following 2-APB treatment, compared to 4.4 + 0.3 a.u for untreated platelets; Figure 4c,d, *p* < 0.05). These findings implicate Zn^2+^ in the regulation of cation influx.

We hypothesized that cation channels involved in SOCE could also facilitate [Zn^2+^]_o_ influx. To test this, platelets were loaded with FZ-3 and stimulated with thrombin or TG, and Zn^2+^ influx was quantified using flow cytometry. In the absence of [Zn^2+^]_o_, thrombin stimulation did not affect FZ-3 fluorescence. However, with 30 μM [Zn^2+^]_o_, thrombin stimulation led to an increase in FZ-3 fluorescence (to 1.7 ± 0.1 a.u for [Zn^2+^]_o_ and 1.5 ± 0.1 a.u for [Zn^2+^]_o_/[Ca2]_o_; Figure 4e,f, *p* < 0.0001) demonstrating an activation-dependent [Zn^2+^]_o_ influx. FZ-3 fluorescence was unaffected by [Ca^2+^]_o_. Further experiments employing the cation chelators BAPTA and TPEN abrogated increases in FZ-3 fluorescence, confirming that Zn^2+^ was responsible for the FZ-3 signal. The pre-treatment of platelets with KB-R or 2-APB had no significant effect on the [Zn^2+^]_i_ signal evoked in the presence of [Zn^2+^]_o_ or [Ca^2+^]_o,_ or a combination of both cations (Figure 4e,f).

TG was used to deplete internal stores, and Zn^2+^ influx into FZ-3-loaded platelets quantified using flow cytometry. In the absence of [Zn^2+^]_o_, TG stimulation did not result in FZ-3 fluorescence in the absence of external cations (Figure 4g,h), and [Ca^2+^]_o_ did not effect FZ-3 responses (1.2 ± 0.1 a.u; Figure 4g,h, ns). However, peak FZ-3 fluorescence increased in response to TG stimulation. F/F_0_ was 3.3 + 0.9 a.u or 2.9 + 0.1 a.u. in the presence of [Zn^2+^]_o_ or [Zn^2+^]_o_/[Ca^2+^]_o_, respectively (Figure 4g,h, *p* < 0.001). TPEN or BAPTA abolished TG-evoked increases in FZ-3 fluorescence (Figure 4g,h). TG-mediated F/F_0_ signals did not differ upon pre-treatment with 30 μM KB-R in comparison to the vehicle control (Figure 4e,f, ns). However, in 2-APB-treated platelets, the Zn^2+^ signal was significantly reduced in the presence of [Zn^2+^]_o_ (2.1 ± 0.1 a.u) compared to the vehicle control (3.3 ± 0.1 a.u; Figure 4g,h, *p* < 0.0001). This was also the case for [Zn^2+^]_o_/Ca^2+^]_o_, where the peak signal was 1.9 ± 0.1 a.u (Figure 4g,h; *p* < 0.001).

These data indicate a mechanism where the depletion of cation stores in platelets following SERCA inhibition results in increased [Zn^2+^]_i_ as a result of Zn^2+^ influx from the extracellular medium in a similar manner to SOCE. We described this mechanism as store-operated Zn^2+^ entry (SOZE).

### 2.4. [Zn^2+^]_o_ Regulates Intracellular Signaling via the Regulation of Zn^2+^ Stores

Zn^2+^ signaling may play a regulatory role in canonical Ca^2+^ signaling responses upon platelet activation. Zn^2+^ regulates Ca^2+^ efflux from platelet stores and it is likely that distinct intracellular Zn^2+^ stores contribute to Ca^2+^-dependent platelet responses [16,17]. In nucleated cells, PLC-dependent generation of IP_3_ causes cation store depletion resulting in STIM1 and Orai1 clustering with a resultant extracellular Ca^2+^ influx by SOCE-mediated activation [29,51]. Exogenous Zn^2+^ induces ZIP7-mediated Zn^2+^ influx leading to tyrosine kinase activation in a process that is regulated by CK2 [50]. We investigated the role of the Zn^2+^ transporter, ZIP7, in elevating [Zn^2+^]_i_ by facilitating Zn^2+^ entry in platelets.

The activation of platelets with CRP-XL (1 µg/mL) or Zn^2+^ (100 µM) led to an increased expression of ZIP7 on the platelet surface demonstrated by flow cytometry (Figure 5a,b; *p* < 0.001). The stimulation of platelets with CRP-XL, but not the TP agonist U46619, resulted in increases in ZIP7 phosphorylation consistent with the role of GPVI in Zn^2+^ signaling (Figure 5c; *p* < 0.001) [14]. CRP-XL- and Zn^2+^-mediated ZIP7 phosphorylation was abrogated by pre-treatment with the CK2 inhibitor CX-4945 (Figure 5c), consistent with the roles of CK2 and ZIP7 in Zn^2+^ release from stores. The role of ZIP7 in Zn^2+^ influx was further investigated using the ZIP7 inhibitor NVS-ZP7-4. The stimulation of FZ-3-loaded platelets with 100 µM Zn^2+^ following NVS-ZP7-4 inhibition (5 μM) resulted in reduced FZ-3 fluorescence in a concentration-dependent manner (Figure 5d, *p* < 0.001). These findings demonstrate the role of ZIP7 in transiently elevating [Zn^2+^]_i_ in response to extracellular signals, and the role of ZIP7 phosphorylation involving CK2.

The relative contributions of [Zn^2+^]_o_ influx or [Zn^2+^]_i_ release from stores to [Zn^2+^]_i_ signals remains unclear. Therefore, we utilized flow cytometry to investigate the influence of Ca^2+^- or Zn^2+^-release mechanisms on activation-dependent [Zn^2+^]_i_ signals in the presence and absence of [Zn^2+^]_o_, to isolate the contribution of [Zn^2+^]i to intracellular stores.

As expected, a higher FZ-3 signal was observed following [Zn^2+^]_o_ stimulation compared to CRP-XL (7375.6 ± 1394.2 a.u. and 3962 ± 328.6, respectively; Figure 6), consistent with contributions to [Zn^2+^]_i_ from both extracellular and intracellular sources, but with a greater relative contribution from [Zn^2+^]_o_. The effects of the inhibitions of IP_3_R (with Teriflunomide, 100 μM) [39], Orai1 (AnCOA4, 5 μM) [52], or ZIP7 (NVS-ZP7-4, 5 μM) were assessed. The inhibition of IP_3_R, Orai1, or ZIP7 reduced Zn^2+^-mediated FZ-3 increases to 2533.4 ± 385.2 a.u., 1549 ± 240.5 a.u., and 1289.4 ± 155.6 a.u. respectively, compared to the vehicle control (7375.6 ± 1394.2 a.u.; Figure 6, *p* < 0.001). CRP-XL stimulation (100 μg/mL) in the absence of extracellular [Zn^2+^]_o_ resulted in FZ-3 signals that were insensitive to the inhibition of IP_3_R, ZIP7, or Orai1 (Figure 6, ns) indicating that Zn^2+^ was not being released from intracellular stores through IP_3_R (as is the case with Ca^2+^ signaling) or ZIP7.

## 3. Discussion

Here, we demonstrate for the first time that Zn^2+^ influx into platelets and subsequent platelet activation are partially blocked by the conventional TRP channel and NCX blockers and an inhibitor of ZIP-7, indicating multiple pathways for Zn^2+^ entry into platelets.

In nucleated cells, [Zn^2+^]_o_ uptake and the elevation of [Zn^2+^]_i_ are mediated by the involvement of Zn^2+^-permeable ion channels, exchangers, and Zn^2+^ transporters, including NCX and TRP channels [20,39,53]. Our data show that KB-R (a reverse-mode NCX inhibitor) caused substantial reductions in ZIPA (Figure 1c), suggesting a distinct role for these exchangers in Zn^2+^ influx into platelets. Reductions in ZIPA were associated with the reduced fluorescence of FZ-3-loaded platelets, consistent with a reduction in reduced [Zn^2+^]_o_-induced Zn^2+^ influx (Figure 2a,b). These observations are consistent with a previous study that shows NCX’s role in mediated Zn^2+^ influx in neuronal cells and that Zn^2+^ allosterically inhibits Na^2+^-independent Ca^2+^ release by inhibiting NCX activity [18,41].

Inhibition of both Zn^2+^ influx and ZIPA were observed following pre-treatment with 2-APB, SKF, or FFA, which suggests the involvement of TRP channels (Figure 2a,b,d). The lack of the full inhibition of Zn^2+^ influx as a result of NCX or TRP inhibitions indicates the existence of alternative Zn^2+^ entry pathways. In addition to being a TRP channel inhibitor, 2-APB also inhibits IP_3_R, and therefore could potentially have inhibited [Ca^2+^]_i_ release from IP_3_-sensitive stores. Among the TRP channels, TRPC1, TRPC3, TRPC4, TRPC5, TRPC6, and TRPV1 are expressed on the platelets [54,55]. Specific inhibition identified roles for TRPC6, TRPC5, and TRPV1 in mediating Zn^2+^ influx (Figure 2b), which is consistent with previous studies that show TRPC6 to be involved in Zn^2+^ influx in nucleated cells [24,56]. Further evidence for the involvement of TRP channels comes from the observation that OAG and GO6983 treatments (both TRP channel activators) resulted in increased Zn^2+^ influx (Figure 2c,d).

We observed Zn^2+^-induced phosphorylation of substrates of Ca^2+^-dependent kinases, PKC, and MLCK (Figure 3). The activation of MLCK requires interactions with the Ca^2+^-binding protein calmodulin (CaM), whilst PKC has a Ca^2+^-binding domain [57,58]. Zn^2+^ was shown to substitute for Ca^2+^ at the EF hand domains of CaM, which may lead to the conformational change and activation of CaM-dependent substrates [47,59]. Additionally, Zn^2+^ binds to and activates PKC isoforms [13,60]. Thus, Zn^2+^ may influence platelet activation by the direct activation of Ca^2+^-dependent proteins. This process may act as a mechanism for priming the platelet for activation in response to low concentrations of conventional agonists [14].

Depletion of Ca^2+^ stores leads to the opening of membrane Ca^2+^ channels in a variety of cell types, including platelets, in a process known as store-dependent Ca^2+^ entry (SOCE). In platelets, the site of the Ca^2+^ store is the DTS. Here, we demonstrate that thrombin stimulation or the depletion of intracellular cation stores using thapsigargin results in increases in [Zn^2+^]_i_ (Figure 4). As this signal was not observed in the absence of subactivatory (30 μM) [Zn^2+^]_o_, we conclude that this signal is due to Zn^2+^ influx, in a process we describe as store-operated Zn^2+^ entry (SOZE). Conversely, thrombin or thapsigargin stimulation results in increases in [Ca^2+^]_i_ in the absence of any extracellular cation, consistent with Ca^2+^ from intracellular stores. Therefore, SOZE does not result in the liberation of Zn^2+^ from intracellular stores.

SOZE, but not SOCE, was sensitive to TPEN pre-treatment, both confirming the nature of the Zn^2+^ signal and providing evidence that SOCE is not dependent on Zn^2+^. As neither SOCE nor SOZE were affected by KB-R, NCX is not involved in these processes. This is consistent with the previous data which indicate that NCX is not involved in the regulation of [Ca^2+^]_i_ in neuronal cells [41]. Interestingly, thapsigargin-, but not thrombin-mediated cation influx, was influenced by 2-APB. As 2-APB is also known to inhibit IP_3_R, this indicates the absence of IP_3_R activity in SOCE [61]. The extent of Ca^2+^ influx in SOCE was reduced in the presence of [Zn^2+^]_o_, consistent with competition for cation entry mechanisms. Conversely, the rate and extent of Zn^2+^ influx were unaffected by [Ca^2+^]_o_, suggesting that Zn^2+^ entry mechanisms, whilst being cation non-specific, favored Zn^2+^. Competition between Ca^2+^ and Zn^2+^ during intestinal absorption has been previously described where the two ions share a common transport pathway [62]. Hence, we suggest the possibility of competition between the two ions occurring via Zn^2+^ entry mechanisms, including TRP channels and NCX. Additionally, inhibition by 2-APB did not affect TG-induced Zn^2+^ influx in the presence of exogenous 2 mM/30 μM [Ca^2+^]_o_/[Zn^2+^]_o_ (Figure 4), suggesting that NCX mediates Zn^2+^ influx, but does not influence store-operated Ca^2+^ release.

SOCE involves the detection of store Ca^2+^ levels by Orai, with subsequent signaling to the membrane cation, STIM1. Whether SOZE is regulated by the detection of Ca^2+^ or Zn^2+^ depletion is not known. Platelets have a Zn^2+^ store that is released into the cytosol following agonist stimulation, in a manner that is consistent with a secondary messenger [14,26]. However, the nature of the Zn^2+^ store and the mechanism by which Zn^2+^ exerts its actions has received little attention. Previous studies indicate that the store is sensitive to platelet stimulation via GPVI and TP, but not PARs, and is sensitive to the redox state [14]. A further mechanism by which cells regulate Zn^2+^ levels is through Zn^2+^ transporters, of which ZIP7 is present in the platelet proteome [63]. In nucleated cells, the activation of tyrosine kinase pathways depend on the release of Zn^2+^ from the ER into the cytosol, mediated by the Zn^2+^ transporter, ZIP7 [64,65,66]. We confirm that ZIP7 is present on the platelet surface (Figure 5) and that platelet activation via Zn^2+^ or CRP-XL leads to increases in surface expression. This is consistent with the presence of ZIP7 on intracellular membranes of the open canalicular system, which becomes externalized upon platelet activation. Platelet activation using Zn^2+^ results in an increase in ZIP7 phosphorylation, which is sensitive to inhibition of CK2, indicating an activatory pathway. These results complement the findings on nucleated cells, where ZIP7 is regulated via CK2-mediated phosphorylation in MCF-7 cells. In this work, Zn^2+^ stimulation resulted in an increase in [Zn^2+^]_I_, which was inhibited upon ZIP7 knockdown via siRNA [65].

Previous work has demonstrated increases in [Zn^2+^]_i_, both as a result of Zn^2+^ influx and release from stores [12,14,26]. To determine the relative contributions of each pathway to [Zn^2+^]_i_ increases, we stimulated FZ-3-loaded platelets with [Zn^2+^]_o_ or CRP-XL (in the absence of [Zn^2+^]_o,_ precluding Zn^2+^ influx). Both pathways result in increases in [Zn^2+^]_i_ with Zn^2+^ stimulation generating a stronger signal, consistent with a greater contribution from Zn^2+^ influx. Other published works have implicated IP_3_R in the persistent, rapid [Zn^2+^]_I_ increases in cortical neurons [67]. To investigate the influence of IP_3_R on Zn^2+^ store release, we inhibited IP_3_R and observed reduced [Zn^2+^]_i_ increases in the response to [Zn^2+^]_o_, but not CRP-XL. This profile is consistent with the roles of IP_3_R, ZIP7, and Orai1 in Zn^2+^ influx, but not for agonist-dependent Zn^2+^ release in the absence of [Zn^2+^]_o_. Therefore, Zn^2+^ is not stored and released in a similar manner to Ca^2+^, and ZIP7 was only effective in Zn^2+^ influx following activation-mediated externalization. This is consistent with the observation that [Zn^2+^]_i_ release, but not Ca^2+^ release, is sensitive to the platelet redox state [14,26] and is supportive of a role for redox-sensitive cation storage proteins, such as metallothioneins. Further experiments are required to further elucidate the nature of the Zn^2+^ store in platelets. The reason why IP_3_R inhibition reduces [Zn^2+^]_o_ influx is unclear; however, it is suggestive of a role for Ca^2+^ in regulating Zn^2+^ entry. This is consistent with the observation that both TG- and thrombin-mediated increases in [Ca^2+^]_i_ were reduced in the presence of [Zn^2+^]_o_ (Figure 4). This observation may suggest cross-talk between Zn^2+^- and Ca^2+^-handling mechanisms during platelet activation.

In conclusion, this study provides evidence that platelets are able to detect and respond to increases in [Zn^2+^]_o_ via influx through TRP channels, the NCX exchanger, and ZIP7 (Figure 7). Store depletion facilitates Zn^2+^ entry via Orai1, which we have termed store-operated Zn^2+^ entry (SOZE). ZIP7 is externalized following platelet activation. Both influx and Zn^2+^ release from stores both contribute to increases in [Zn^2+^]_i_. It is likely that other cell systems employ similar mechanisms to detect and respond to changes in [Zn^2+^]_o_.

## 4. Materials and Methods

Materials: 2-Aminoethoxydiphenyl borate (2-APB), SKF 96365 hydrochloride (SKF; 1-[2-(4-Methoxyphenyl)-2-[3-(4-methoxyphenyl)propoxy]ethyl-1H-imidazole hydrochloride), KB-R7943 mesylate (KB-R; 2-[2-[4-(4-Nitrobenzyloxy)phenyl]ethyl]isothiourea mesylate), SN-6 (2-[[4-[(4-Nitrophenyl)methoxy]phenyl]methyl]-4-thiazolidinecarboxylic acid ethyl ester), SAR7334 (4-[[(1R,2R)-2-[(3R)-3-Amino-1-piperidinyl]-2,3-dihydro-1H-inden-1-yl] oxy]-3-chlorobenzonitrile dihydrochloride), 9-Phenanthrol, AM0902 (1-[[3-[2-(4-Chlorophenyl)ethyl]-1,2,4-oxadiazol-5-yl]methyl]-1,7-dihydro-7-methyl-6H-purin-6-one), Teriflunomide (2-Cyano-3-hydroxy-N-[4-(trifluoromethyl)phenyl]-2-butenamide), and integrilin were from Tocris (Bristol, UK). Flufenamic acid (FFA), 1-Oleoyl-2-acetyl-sn-glycerol (OAG), thrombin, calcium chloride, and zinc sulphate were supplied by Sigma (Poole, UK), and thapsigargin (TG) was from Calbiochem (Nottingham, UK). AC1903 and NVS-ZP7-4 were from MedChem express (Monmouth Junction, NJ, USA); AMG9810 ((2E)-N-(2,3-Dihydro-1,4-benzodioxin-6-yl)-3-[4-(1,1-dimethylethyl)phenyl]-2-propenamide) was from Abcam (Cambridge, UK). AnCoA4 was from Merck (Watford, UK). CX-4945 was from Stratetech, Ely, UK). Unless stated, all other reagents were from Sigma Aldrich.

Preparation of washed human platelets: ethical approval for this study was obtained from the Faculty Research Ethics Committee at Anglia Ruskin University. Human blood was collected from healthy volunteers who had not taken medication for two weeks, following informed consent in accordance with the Declaration of Helsinki. Blood was collected in 11 mM of sodium citrate and washed platelets were prepared as described previously [12]. For studies of Ca^2+^ or Zn^2+^ mobilizations, aliquots of platelet-rich plasma (PRP) were retained. Platelets were resuspended using a calcium-free Tyrodes buffer (CFT) containing, in mM, 140 NaCl, 5 KCl, 10 HEPES, 5 Glucose, 0.42 NaH_2_PO_4_, and 12 NaHCO_3_, titrated to pH 7.4 with NaOH.

Light transmission aggregometry: platelet aggregation was monitored as described previously using an AggRam light transmission aggregometer (Helena Biosciences, Gateshead, UK) [12]. Washed platelet suspensions in CFT were stimulated under stirring conditions at 37 °C and aggregation traces were acquired digitally using proprietary software (HemoRam v1.3, Helena Biosciences). Platelets were stimulated by 100 µM ZnSO_4_ following pre-incubation (up to 15 min) with specified channel blockers or vehicle control (0.1% DMSO).

Fluorometry: platelets in PRP were loaded with Fluozin-3 (FZ-3, 1 µM, Invitrogen, Paisley, UK) or Fluo-4 (1 µM, Invitrogen) for 30 min, at 37 °C. Platelets were collected by centrifugation (350× *g*, 15 min), resuspended in CFT, and rested at 37 °C for 30 min prior to use. FZ-3 fluorescence data were acquired under 494 nm excitation and 516 nm emission using a Fluoroskan Ascent Fluorometer (ThermoScientific, Oxford, UK).

Flow cytometry: PRP was loaded with Fluo-4 or FZ-3 (1 µM) and elevations of [Ca^2+^]_i_ or [Zn^2+^]_i_ were monitored over a 4 min period using an Accuri C6 flow cytometer (BD Biosciences, Oxford, UK). Platelets were stimulated by thrombin (1 U/mL) in the presence of 2 mM [Ca^2+^]_o_, 30 μM [Zn^2+^]_o_, or a combination of both. Where stated, platelets were incubated with 1 µM TG, a SERCA pump inhibitor, for 5 min in the absence of [Ca^2+^]_o_ to assess Ca^2+^ release from the DTS. For experiments assessing the role of NCX or TRP channels, platelets were incubated with inhibitors for 30 min at 37 °C, before stimulation with 100 µM Zn^2+^, and the FZ-3 signal was measured for 5 min. For experiments assessing [Zn^2+^] release from internal stores, platelets were pre-treated with inhibitors against Orai1, IP_3_R, or ZIP7 for 30 min at 37 °C, prior to stimulation with 100 µM Zn^2+^ or 1 μg/mL CRP-XL (Cambcol, Ely, UK).

Western blotting: Western blotting was performed as described previously [12]. Briefly, washed platelet suspensions were pre-treated with the α_IIb_β_3_ inhibitor integrilin (5 μM) for 5 min prior to stimulation with 30 μM or 1 mM Zn^2+^ for the given time periods, where platelets were then lysed with RIPA buffer (150 mM NaCl, 50 mM Tris-HCL, pH 8.0, 1% nonidet P-40, 0.5% Na deoxycholate, 0.15 SDS). Lysates were separated on 8% SDS-PAGE or 4–12% Gradient NuPAGE Bis-Tris (BioRad) gels and transferred to PVDF membranes. Zn^2+^-induced protein phosphorylation was assessed using antibodies against phospho-β3 and Y759 (AbCam, Cambridge, UK), PKC substrate (AbCam), or p-MLC S19 (Cell Signalling Technology, Danvers, MA, USA), or anti-Phospho-ZIP7 antibodies (Cambridge Bioscience, Cambridge, UK), followed by anti-mouse, anti-rabbit, or anti-goat HRP-conjugated secondary antibodies (AbCam. Antibodies against total β3 or β-actin (Santa Cruz, Heidelberg, Germany) were included as loading controls. Blots were representative of 3–4 platelet preparations.

Data analysis and statistics: maximum aggregation and mean fluorescence values were calculated using Microsoft Excel 365 (Microsoft, Redmond, WA, USA). Representative traces of FZ-3 or Fluo-4 fluorescence responses were generated using Flow Jo (V10.2, Ashland, OR, USA). Western blots were analyzed using ImageJ (v1.45, NIH, Bethesda, MD, USA). Data were analyzed in GraphPad Prism by a two-way ANOVA of Student’s *t*-test, where *p* < 0.0001 (****), *p* < 0.001 (***), *p* < 0.01 (**), *p* < 0.05 (*), and *p* > 0.05 (ns) were indicated.

## Figures and Tables

**Figure 1 ijms-24-11689-f001:**
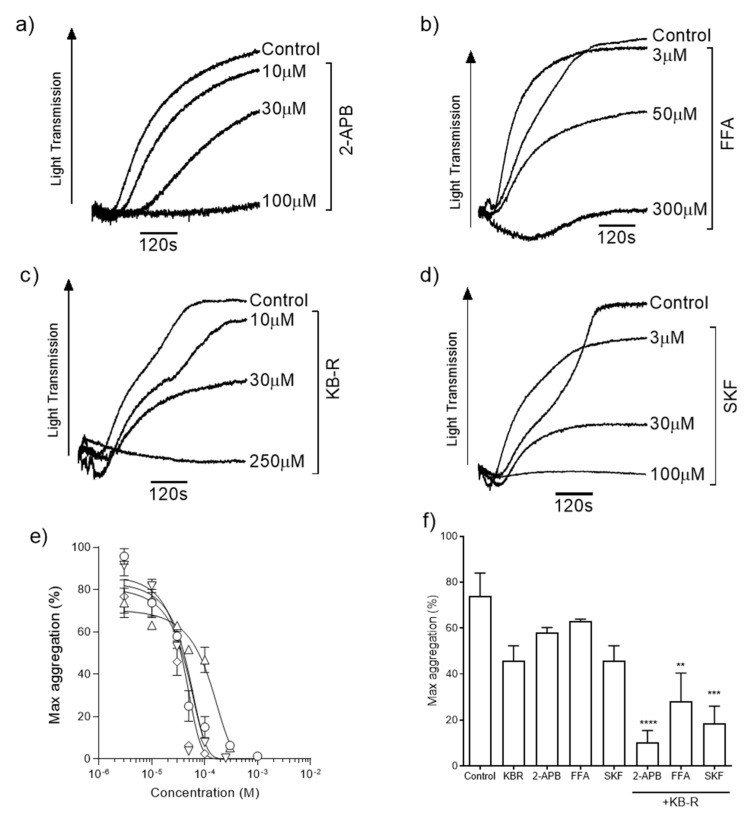
Inhibition of Zn^2+^-induced aggregation by TRP channel and NCX blockers. Representative aggregation traces for washed platelets stimulated with Zn^2+^ in the presence of ion channel inhibitors 30 µM 2-APB (**a**), 30 µM FFA (**b**), 30 µM KB-R (**c**), and 30 µM SKF (**d**). Concentration–response relationships were calculated for each reagent (**e**) and IC_50_ values are reported in Table 1. Co-application of 30 µM KB-R plus 30 µM 2-APB, 30 µM FFA, or 30 µM SKF caused a further inhibition of platelet aggregation relative to the vehicle control (**f**). Experiments were performed in the absence of [Ca^2+^]_o_. Data are the means ± SEM of independent experiments. *p* < 0.0001 (****), *p* < 0.001 (***), and *p* < 0.01 (**) are indicated.

**Figure 2 ijms-24-11689-f002:**
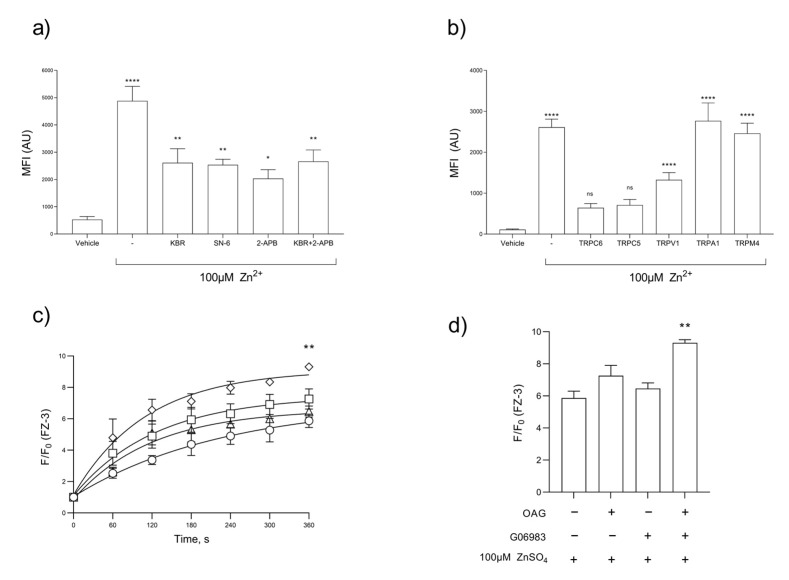
Platelet Zn^2+^ entry is mediated by TRP channel and NCX-mediated pathways. FZ-3-loaded washed platelet suspensions were stimulated by 100 µM Zn^2+^ and changes in fluorescence in response to the inhibition of NCX-mediated pathways were observed by flow cytometry (**a**,**b**) or fluorometry (**c**,**d**) and (**b**,**d**). (**a**) Inhibitors included 2-APB (30 µM), KB-R (30 µM), SN6 (0.5 µM), or a combination of 2-APB and KB-R (30 µM each), in comparison to the vehicle control or unstimulated platelets. (**b**) Inhibition of TRPC6 (SAR7334, 15 µM), TRPC5 (AC1903**,** 15 µM), TRPV1 (AMG 9810, 2.5 µM), TRPA1 (AM0902, 15 µM), and TRPM4 (9-phenanthrol, 15 µM), compared to the vehicle control or unstimulated platelets. (**c**) FZ-3-loaded platelet suspensions were stimulated with Zn^2+^ (○, 100 μM), following pre-treatment with OAG (□, TrpC6 stimulator, 100 μM), or GO6983 (△, PKC inhibitor that enhances TrpC6 activity, 300 nM) or both OAG and GO6983 in combination (♢). Maximum FZ-3 fluorescence was calculated (**d**). Data are the means ± SEM of a minimum of 4 independent experiments. *p* < 0.0001 (****), *p <* 0.01 (**), *p <* 0.05 (*), and *p >* 0.05 (ns) are indicated.

**Figure 3 ijms-24-11689-f003:**
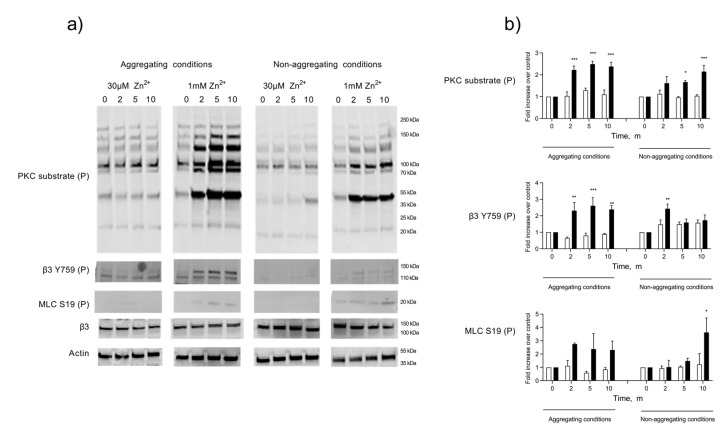
Zn^2+^-mediated platelet activation induces the phosphorylation of Ca^2+^-dependent substrates. Washed platelets were stimulated by 30 µM or 1 mM Zn^2+^ in an aggregometer under stirring conditions, following pre-treatment with integrilin (5 µM, non-aggregating conditions), or untreated (aggregating conditions). Whole-cell lysates were generated at 0, 2, 5, and 10 min. Phosphorylations of PKC substrates, MLC, and β3 were analyzed by Western blotting (**a**). Samples were probed for β3 or actin to assess equal protein loading. Densitometric analyses of selected bands (MLC S19P and β3 Y759P) or whole lanes (PKC substrate) were performed to quantify changes in phosphorylation relative to the control at each time point (**b**). White bars: 30 µM Zn^2+^, black bars: 1 mM Zn^2+^. Data are the means ± SEM of a minimum of four independent experiments. *p* < 0.001 (***), *p* < 0.01 (**), and *p* < 0.05 (*) are indicated.

**Figure 4 ijms-24-11689-f004:**
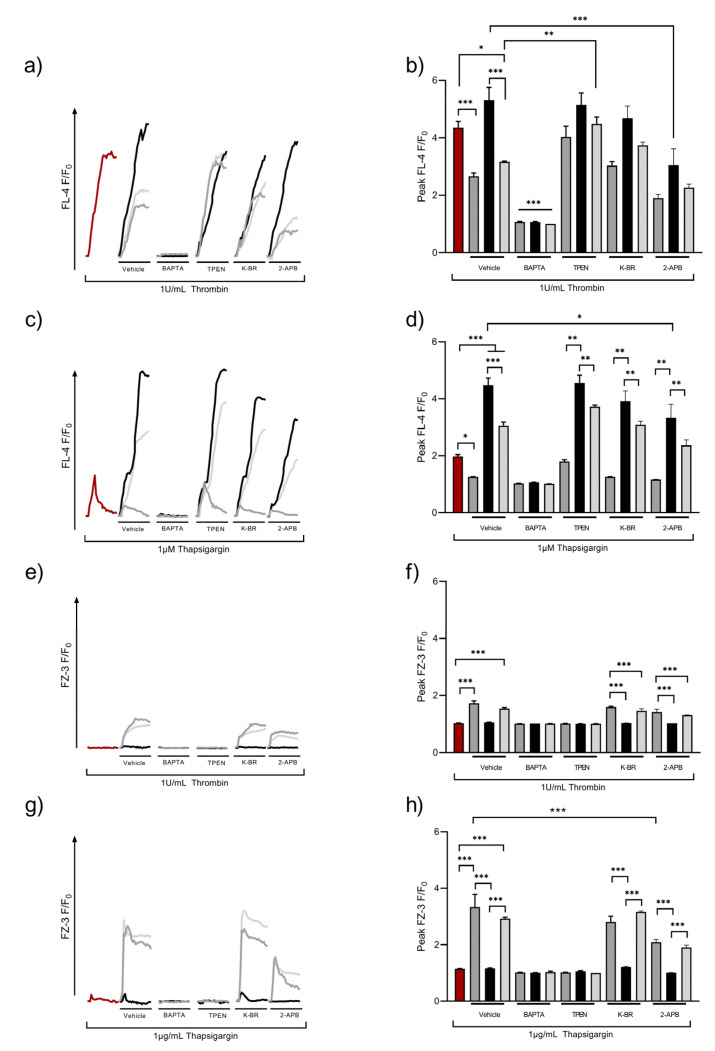
Zn^2+^ influx is regulated by agonist- and store-operated mechanisms. Fluo-4 (FL-4)- or FZ-3-loaded platelet suspensions were stimulated after 30 s by 1 U/mL thrombin (**a**,**b**,**e**,**f**) or 1 μM thapsigargin (**c**,**d**,**g**,**h**), either in the absence of external cations (red line) or in the presence of 30 µM Zn^2+^ (dark gray), 2 mM Ca^2+^ (black), or both 2 mM Ca^2+^/30 μM Zn^2+^ (light gray), and changes in fluorescence were observed by fluorometry. Representative traces (**a**,**c**) following thrombin or thapsigargin stimulations under control conditions (no cation applied), in addition to pre-treatment with the vehicle control (dH_2_O), BAPTA (10 μM), TPEN (25 μM), 2-APB (30 μM), or KB-R (30 μM) to assess the mechanisms involved in mediating Zn^2+^ influx upon activation are shown. Peak responses are also shown (**b**,**d**–**f**). Data are the means ± SEM of a minimum of 4 independent experiments. *p* < 0.001 (***), *p* < 0.01 (**), and *p* < 0.05 (*) are indicated.

**Figure 5 ijms-24-11689-f005:**
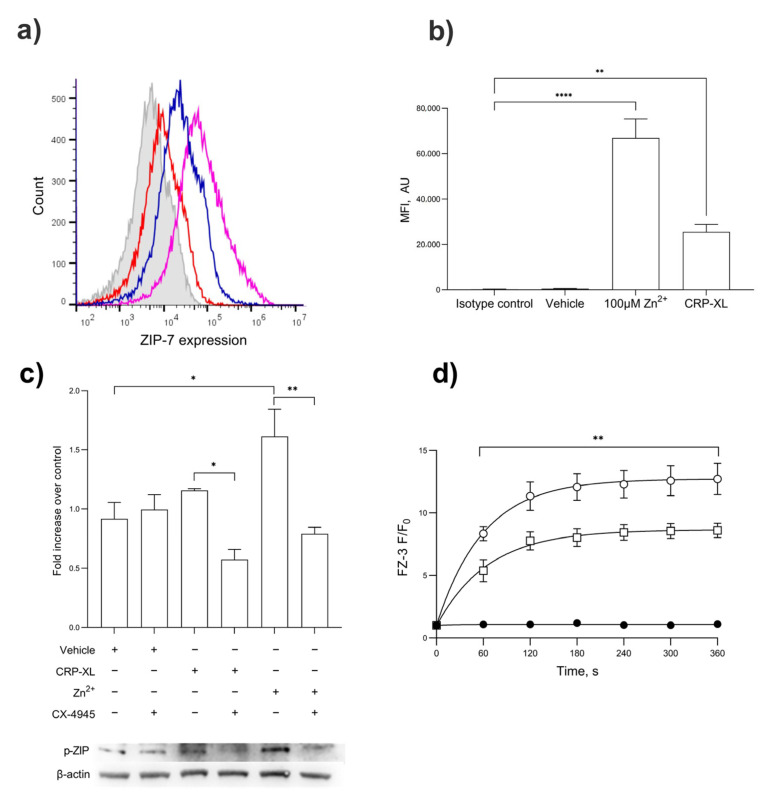
ZIP7 is externalized upon platelet activation, where it contributes to Zn^2+^ influx. (**a**) Flow cytometry demonstrates the presence of ZIP7 on the surface of platelets. ZIP7 expression was demonstrated using an anti-ZIP7 antibody in unstimulated platelets (red) or in platelet suspensions stimulated by CRP-XL (1 μg/mL, blue) or Zn^2+^ (100 μM, purple), compared to the isotype control (gray). (**b**) Quantitation of mean fluorescent intensity of ZIP7 expression on platelets treated with Zn^2+^ (100 µM) or CRP-XL (1 µg/mL) compared to unstimulated or isotype controls. (**c**) Western blotting for phospho-ZIP7 in platelets stimulated with CRP-XL (1 µg/mL) or ZnSO_4_ 100 µM) following pre-treatment with the casein kinase II inhibitor CX-4945. (**d**) Fluorescence in FZ-3-loaded platelets was measured in response to 100 µM Zn^2+^ by fluorimetry, following pre-treatment with NVS-ZP7-4 (10 µM, □) or vehicle control (DMSO, ○), or unstimulated (●) Data are representative of a minimum of 4–6 independent experiments. *p* < 0.01 (**), *p* < 0.05 (*), and *p* < 0.0001 (****), are indicated. Error bars indicate the standard error of the mean (SEM).

**Figure 6 ijms-24-11689-f006:**
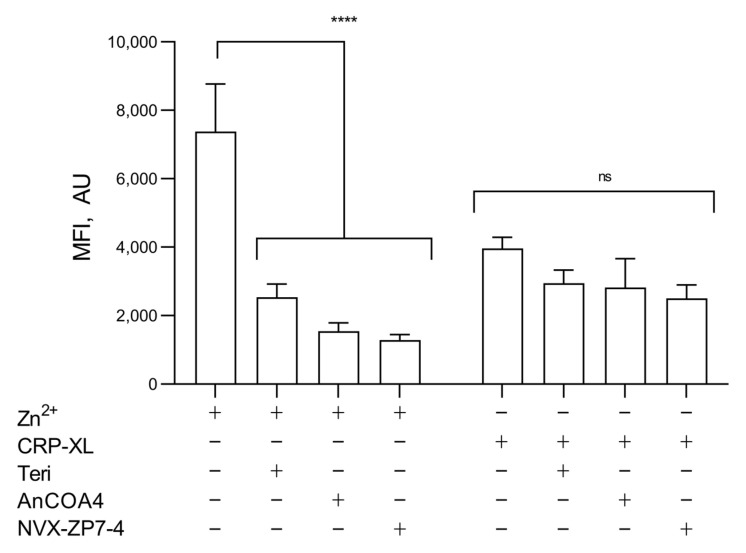
Both Zn^2+^ influx and agonist-dependent release from stores contribute to increases in [Zn^2+^]_i_. FZ-3-loaded platelets were stimulated with CRP-XL (1 µg/mL) or [Zn^2+^]_o_ (100 µM) following pre-treatment with inhibitors of IP_3_R: Teriflunomide (Teri, 100 µM), Orai1: AnCOA4 (5 μM), and ZIP7: NVS-ZP7 (5 μM), and fluctuations in fluorescence were recorded using fluorometry. Data are means ± SEM of a minimum of 4–6 independent experiments. *p* < 0.0001 (****), *p* > 0.05 (ns) are indicated.

**Figure 7 ijms-24-11689-f007:**
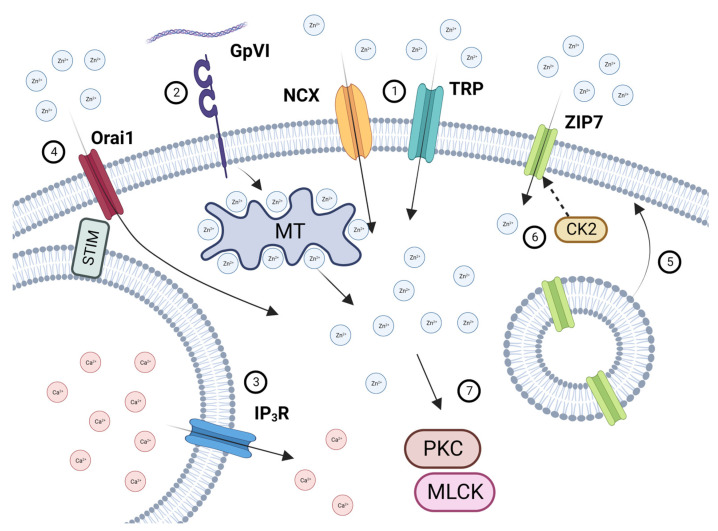
Schematic illustration of [Zn^2+^]_o_ and [Zn^2+^]_I_ handling in platelets. Increases in [Zn^2+^]_o_ following endothelial damage, atherosclerotic plaque rupture, or platelet degranulation results in Zn^2+^ influx via NCX and TRP channels (1). Platelet activation by GPVI (2) results in the liberation of [Zn^2+^]_i_ from intracellular stores likely to be redox-sensitive proteins, such as metallothioneins). Platelet activation also results in Ca^2+^ store depletion (3), which is detected by STIM, leading to [Zn^2+^]_o_ influx via Orai1 (4). Activation-dependent degranulation leads to the externalization of ZIP7 on the platelet surface (5), which is phosphorylated by CK2, further facilitating Zn^2+^ influx (6). Increases in [Zn^2+^]_i_ results in the activation of PKC and MLCK, contributing to downstream platelet activatory responses (7). Created with BioRender.com.

**Table 1 ijms-24-11689-t001:** IC_50_ values for the inhibition of 100 µM Zn^2+^-induced platelet aggregation.

Drug	Target	IC_50_
2-APB	TRP channels/SOCE	34.5 ± 3.0 µM
FFA	TRP channels	76.7 ± 9.8 µM
KB-R	Reverse-mode NCX	28.9 ± 1.3 µM
SKF	TRP channels	23.9 ± 2.5 µM

## Data Availability

The data presented in this study are openly available in FigShare at https://doi.org/10.25411/aru.23703288.v1.

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
