# Peer review of "Delineating Zinc Influx Mechanisms during Platelet Activation"

_ijms, 2023, doi:10.3390/ijms241411689_

Round 1

Reviewer 1 Report

In the presented manuscript entitled "Delineating Zinc Influx Mechanisms During Platelet Activation," Kuravi et al. propose a new Zn2+-dependent signaling pathway in platelets upon Zn2+-induced platelet activation (ZIPA). This mechanism is regulated by ZIP7 and TRP channel-mediated Zn2+ influx and the consequent activation of zinc-sensitive signaling pathways. Extracellular Zn2+ influx is also coupled to store-operated Ca2+ entry, and this new Zn2+ uptake mechanism is called store-operated zinc entry (SOZE). To understand the effects of platelet Zn2+ on this signaling route, the authors used different Zn2+ chelators (TPEN) or Zn2+ supplements (ZnSO4), as well as TRP channel blockers in different experimental settings, and showed that TRP channels contribute to Zn2+ influx, but an alternative pathway still exists, which is probably regulated by ZIP7 or other ZIP isoforms in platelets. The aim of the study was well-defined and raised several important questions in the research field of thrombosis and hemostasis. Investigating the molecular mechanisms of platelet Zn2+ uptake and release is clinically relevant and essential in many human diseases, such as storage pool diseases, in which platelet Zn2+ homeostasis can be defective. Furthermore, their experimental evidence indicates that GPVI activation with CRP induces accumulation of ZIP7 in the plasma membrane, and Zn2+ influx is activated by CK2-mediated phosphorylation of ZIP7.

I find that the presented manuscript is important in better understanding the contribution of Zn2+ -mediated signaling in platelet activation. Platelet Zn2+ is stored in secretory granules and covalently linked to metallothionein in the cytoplasm. Therefore, degranulation results in Zn2+ deficiency in platelets. According to the authors' data, Zn2+ influx seems to be regulated through ZIP7 and non-selective TRP channels during degranulation and SOCE, probably taking back the extracellular free Zn2+ which is released from granules. This autocrine, Zn2+ -dependent feed-forward loop can further enhance Zn2+-sensitive signaling, probably regulated by PKC or other enzymes, and accelerate platelet activation and aggregation responses.

Specific comments and suggestions:

1.    TRPM7 is a non-selective cation channel that regulates the entry of zinc, magnesium, and calcium in many cell types, including megakaryocytes and platelets. 2-APB is a non-selective inhibitor of SOCE and also inhibits TRPM7 channel activity. Therefore, it would be important to use a more selective TRPM7 channel inhibitor, such as NS8593, in platelets and study the contribution of TRPM7 in ZIPA and SOZE.

2.    Platelet activation is controlled by the AC-PKA-VASP signaling pathway. Zinc supplementation may inhibit this pathway, therefore strongly potentiating integrin activation, and consequent platelet aggregation. It would be important to investigate whether abnormal VASP phosphorylation may be contributed to ZIPA and SOZE.

3.    Previous studies have shown that overexpression of ZIP7 in the presence of Zn2+ supplements regulates the phosphorylation of many enzymes in MCF cells. Interestingly, Lyn, Fyn, and Src kinases are hyper-phosphorylated in MCF cells. To confirm this data in platelets, it would be important to investigate whether Zn2+ supplementation enhances the phosphorylation of Lyn, Fyn, Src and SYK kinases, thereby accelerating GPVI-mediated platelet activation.

4.    To apply the presented results to clinical settings, it would be important to investigate whether platelets isolated from patients with storage pool diseases (SPD) have abnormal ZIPA and SOZE. Defective Zn2+ storage and reduced Zn2+ levels in the cytoplasm of SPD platelets may enhance ZIP7 externalization and consequent Zn2+ influx through the platelet plasma membrane.

5.    The western blots of Fig. 5 C and D are not conclusive. Better quality of western blots with statistics should be provided and the original uncropped blots with protein molecular weight markers should be presented as Supplemental figures.

Author Response

We thank Reviewer 1 for their thoughtful comments.

  1. Many thanks to the reviewers for identifying TrpM7 as a potential Zn entry mechanism. We had not investigated TRPM7 specifically in the manuscript, but it is certainly a target which we would be interested in investigating in the future. We will be conducting further experiments using this inhibitor, however, its omission here does not affect the conclusions of our paper, in that Our conclusions identify a number of Trp channels as a potential Zn entry mechanisms,
  2. We have conducted these experiments and indeed shown that VASP is phosphorylated in response to increases in [Zn2+]I in a published paper (Ahmed et al., 2019), and therefore it would not be appropriate to repeat these experiments here.
  3. This is an interesting observation, which we will certainly be following up on. However, we feel that in the context of the manuscript, experiments to look at Lyn, Fyn and Sec phosphorylation fall beyond the scope of the current work . We will indeed be looking at these in the future.
  4. This is a great suggestion, and one that we will be investigating in the future. However, we feel that it is beyond the scope of the experiments conducted in the current manuscript, which establish the basis on which further clinical studies could be performed.
  5. We have repeated the Western blots with full quantitation. The data are included in the submission. The full blots are provided in the submission.

Reviewer 2 Report

Zinc deficiency results in a prolonged bleeding and can be reversed by dietary supplementation. However, the mechanisms of zinc influx in platelets are still not clear. In this manuscript Kuravi et al identified some mechanisms of zinc transport inside the platelets. However, there are still some questions that could be addressed in the manuscript. First and most important is whether the described mechanisms are physiologically relevant. Free zinc concentration in the blood is around 0.5µM, zinc will be released during platelet activation and elevated zinc will bind to plasma proteins, therefore it hardly will achieve 30, 100 and 1000µM concentrations used in this study. The authors should explain why they used these concentrations. For aggregation experiments and intracellular zinc concentration they used 100 µM, for Western blot 30 µM and 1 mM. In Western blot data presented at Fig. 3 only 1mM! of zinc induces strong PKC activation and MLC phosphorylation. It means that even relatively high (30µM) concentration did not activate these pathways in platelets. Physiological relevance of the presented results should be addressed in the discussion.

Minor.  

1.       Fig. 1. It is not indicated whether the aggregation experiments were done in the presence of calcium or in Ca free solution. This is important because the inhibitors could reduce Ca signal.

2.       2.2. “Zn2+-Induced Platelet Activation is Associated with Phosphorylation of Ca2+-Dependent

Substrates.” Ca2+-Dependent Substrates is not correct. It should be changed to Protein kinases activated by calcium.

3.       Fig. 7C. P-ZIP7 blot looks very strange. Control is not presented and the first lane run bellow the rest lanes. This blot could not quantified and should be repeated. 

Author Response

  1. Many thanks for this observation. Aggregation was done in calcium-free tyrodes buffer.  We have added further information in the figure legend to clarify this
  2. We are happy to make this change, and agree that the reviewers suggestion is a better description of the process
  3. We have repeated the Western Blots (Fig 5) and provided quantification.

Reviewer 3 Report

In this manuscript (ID# ijms-2414323), entitled “Delineating zinc influx mechanisms during platelet activation”, authors Kuravi et al studied the mechanisms of Zinc influx in platelet. Their results showed that TRP channels, NCX exchanger, ZIP7, Orai1, IP3R were all involved in Zinc elevation during platelet elevation. However, the experimental design is not rigorous. The results from this study is not reliable. Several concerns have been listed in the following paragraphs:

1. The results of this study indicate that the zinc transport pathways are so similar with calcium transport pathway. Is it possible that the zinc influx and zinc release observed in this study are mediated by an indirect regulatory effect of calcium?  These results were measured in a physiological solution or in a calcium-free solution?

2. Data in Figure 1 demonstrated that TRP channel and NCX blockers inhibited Zinc-induced platelet aggregation. The evidence from this figure and others is not enough to conclude that Zinc influx is mediated by transportation though those transporters and channels. Therefore, the information in Figure 7 is misleading. The scenario observed could be caused by indirect effect of sodium/calcium transportation and membrane potential drive. How about the effect of ZIP inhibitor?

3. Many studies have demonstrated that Zinc is an inhibitor of TRP channel. This conflict issue should be addressed. In addition, 2-APB, FFA, and SKF block Zinc influx through TRP channel?

4. In Figure3, PKC and MLC phosphorylation assay, the ratio of phosphorylated protein vs total protein should be calculated. Similar issue for Figure 5 C and D 

5. Is STIM1 zinc-sensitive? Dose CRP-XL induce STIM1 and Orai1 cluster formation in platelets?

English language is fine

Author Response

  1. The results were predominantly obtained in washed platelet solutions in calcium-free tyrodes buffer, and were therefore Ca2+ free (this information is included in the methods section)
  2. We agree that data in figure 1 are not conclusive on the mechanism of Zn2+ entry. However, subsequent experiments in figure 2 provide more information here, identifying specific Trp channels (Trp5 and Trp6) as being responsible for Zn2+ entry, in addition to the NCX exchanged. The picture here is complicated, in that there appear to be a number of potential entry routes, and there are likely to be others that we have not discovered.  Experiments with the Zip inhibitor are discussed in Figure 6, where we show that Zn2+ entry is partially reduced following inhibition with the Zip inhibitor, again consistent with our model that multiple entry mechanisms are responsible for Zn2+ entry. We have added further text in the conclusions to clarify this.
  3. Indeed some research has indicated that Zn2+ inhibits Trp channels, and this is also may be apparent in our experiments where SOCE in the presence of external Zn2+ is abrogated. We have added further text in the paper to address this. I am somewhat confused by the reviewers next comment – indeed 2-APB, FFA and SKF are well known to block Zn influx through Trp channels, which is the reason that we used them in our experiments. Here, we show that use of these inhibitors do indeed block Zn2+ entry, which forms the basis of the conclusions to our work. I apologise if I have misunderstood the reviewer, and would be happy to respond to further clarification
  4. We have provided further quantitation for the Western blotting data as suggested by review 2. We are unsure of the reviewers request here in relation to calculating ratios of phosphorylated protein to total protein. In calculating the fold increase over control (fig 3), we are effectively quantifying the increases in phosphorylation following platelet activation, which is a widely used approach to investigating protein phosphorylation. I do not understand how this is different to what the reviewer is suggesting. Apologies if I am not understanding this well, and I would be happy to respond to further clarification.
  5. We have no knowledge of STIM1 being Zn2+ sensitive, nor is there any information that CRP induces STIM1/Orai1 cluster formation in platelets. However, whilst these are interesting lines of investigation, they fall beyond the scope of the current work. We will however consider them in future work.

Round 2

Reviewer 3 Report

no additional recommendation 

no additional recommendation